# SuD-CoTAN: Sulcal Depth-guided Anatomically Consistent Fetal Cortical Surface Reconstruction

**Irina Grigorescu**[1,2] ID                    IRINA.GRIGORESCU@KCL.AC.UK
**Jiaxin Xiao**[2]                              JIAXIN.1.XIAO@KCL.AC.UK
**Yourong Guo**[2]                              YOURONG.GUO@KCL.AC.UK
**Vanessa Kyriakopoulou**[1]                    VANESSA.KYRIAKOPOULOU@KCL.AC.UK
**Alena Uus**[1,2]                              ALENA.UUS@KCL.AC.UK
**Vyacheslav Karolis**[1,2]                     SLAVA.KAROLIS@KCL.AC.UK
**Kaili Liang**[2]                              KAILI.LIANG@KCL.AC.UK
**Mohamed A. Suliman**[2]                       MOHAMED.SULIMAN@KCL.AC.UK
**Qiang Ma**[3]                                 Q.MA20@IMPERIAL.AC.UK
**Daniel Rueckert**[3]                          D.RUECKERT@IMPERIAL.AC.UK
**Bernhard Kainz**[3]                           B.KAINZ@IMPERIAL.AC.UK
**A. David Edwards**[1]                         AD.EDWARDS@KCL.AC.UK
**Joseph V. Hajnal**[1]                         JO.HAJNAL@KCL.AC.UK
**Mary Rutherford**[1]                          MARY.RUTHERFORD@KCL.AC.UK
**Maria Deprez**[*1,2]                          MARIA.DEPREZ@KCL.AC.UK
**Emma C. Robinson**[*1,2]                      EMMA.ROBINSON@KCL.AC.UK

[1] *Department of Early Life Imaging, School of Biomedical Engineering & Imaging Sciences, King's College London, London, United Kingdom*

[2] *Department of Biomedical Computing, School of Biomedical Engineering & Imaging Sciences, King's College London, London, United Kingdom*

[3] *BioMedIA, Department of Computing, Imperial College London, London, UK*

**Editors:** Accepted for publication at MIDL 2026

## Abstract

Accurate and anatomically consistent fetal cortical surface reconstruction is essential for studying early brain development, yet existing methods often lack reliable vertex-wise correspondence and fail to harmonise their outputs across heterogeneous magnetic resonance imaging (MRI) datasets. We introduce Sulcal Depth-guided CoTAN (SuD-CoTAN), a learning-based framework that fits anatomically and topologically consistent cortical meshes directly to T2-weighted MRI and performs alignment to age-matched templates in one single step. All models are trained exclusively on normative samples from the developing Human Connectome Project (dHCP) and evaluated within-sample and on a different acquisition protocol. Results show that SuD-CoTAN generalises to new datasets in ways that harmonise global morphometric properties by better capturing the surface geometry of individual cases; its template fitting is precise, delivering vertex-wise anatomical correspondences that result in sharp weekly averages of sulcal depth and curvature maps in template space. This supports direct vertex-wise Gaussian Process regression of neurodevelopmental trends without a need for any additional registration. Collectively, this whole pipeline runs in ∼3 seconds. This suggests that SuD-CoTAN offers promise as a screening tool for cortical malformations during fetal development.

**Keywords:** fetal MRI, cortical surface reconstruction, deep learning

*  Contributed equally

## 1. Introduction

Characterising fetal cortical development is central to detecting conditions where the folding process deviates from typical trajectories. In the fetal period, disruptions to cortical folding are associated with a range of neurodevelopmental disorders, making precise characterisation of surface geometry critical for quantifying typical and atypical maturation (Garcia et al., 2025; Story et al., 2021; Kyriakopoulou et al., 2014). However, obtaining surfaces that are both anatomically meaningful and harmonised across magnetic resonance imaging (MRI) protocols remains challenging, particularly in fetal MRI.

Current approaches for surface extraction often rely on deformation of the template surface mesh in order to preserve topology of the cortical surfaces (Schuh et al., 2017; Zöllei et al., 2020; Ma et al., 2022) or implicit representations (Cruz et al., 2021; Wang et al., 2023; Gopinath et al., 2023); however, these approaches cannot guarantee vertex-wise anatomical correspondences across the population. These correspondences need to be estimated post-hoc, using spherical projection and registration (Robinson et al., 2014, 2018; Besenczi et al., 2024), to enable accurate detection of subtle cortical abnormalities through high-resolution (vertex-level) comparisons. Additionally, changes in scanner and reconstruction protocols translate to differences in apparent resolution, contrast, and partial volume, resulting in inconsistent cortical surface measures across different acquisition protocols. Classical neonatal pipelines (Makropoulos et al., 2018) can produce high-quality surfaces for neonatal MRI, yet they fail to generalise to fetal scans, where lower resolution leads to partial-volume effects that blur fine cortical structures (see Figure 1).

Current best practice requires lengthy processing, involving several error-prone steps, including intensity-based tissue segmentation (Scott et al., 2011; Gholipour et al., 2017; Makropoulos et al., 2018; Uus et al., 2021), deformable mesh fitting at the boundaries, inflation (for visualisation), and projection to the sphere followed by surface registration (Robinson et al., 2014, 2018; Besenczi et al., 2024). This makes them highly non-practical for use in the clinic. Ideally, surface reconstruction should instead be robust: generating anatomically and topologically correct meshes, that are harmonised across acquisition protocols and automatically aligned to a normative reference cohort for precision detection of outliers.

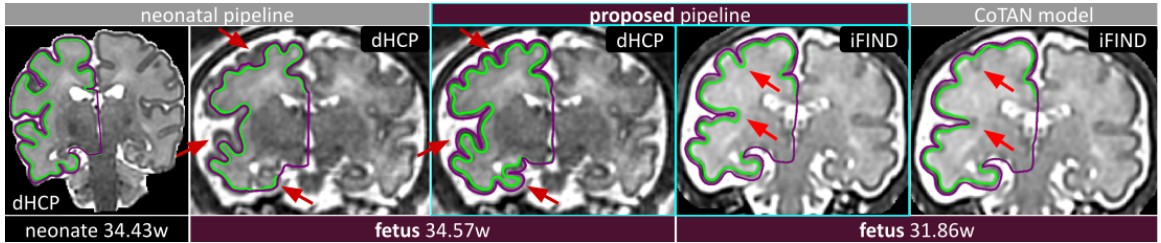

Figure 1: Traditional neonatal surface-reconstruction pipelines do not perform well on fetal MRI (see red arrows), while our proposed SuD-CoTAN framework is able to produce anatomically plausible surfaces for fetal scans from both dHCP and iFIND. The CoTAN (baseline) deep learning models (see Section 3.5) underperform on the unseen iFIND dataset.

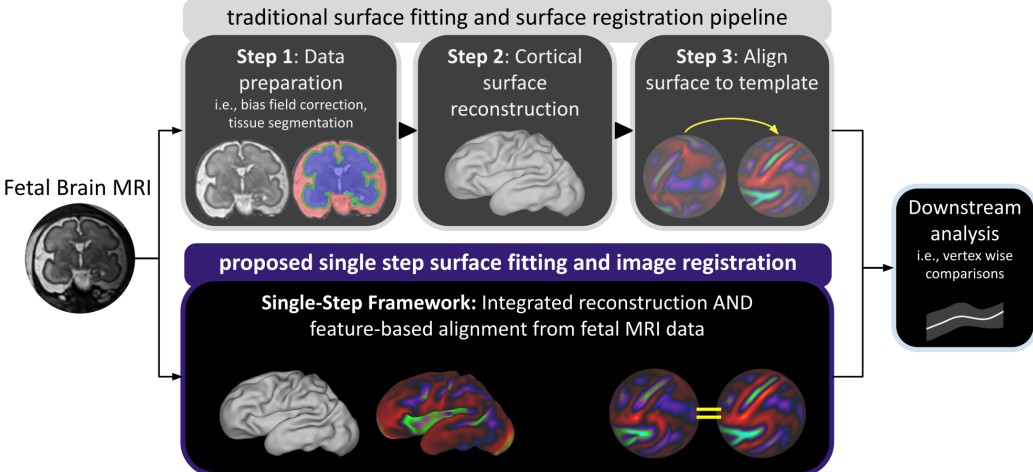

Figure 2: Traditional pipelines (top row) rely on several separate image processing steps: tissue segmentation, followed by surface fitting, followed by spherical alignment to a population-average template for vertex-wise analysis. By contrast, our method (bottom row) integrates template alignment directly into the reconstruction process, producing analysis-ready, template-aligned surfaces directly from raw fetal MRI in a single step.

**Contributions:** We introduce **Sulcal Depth-guided CoTAN (SuD-CoTAN)**, a learning-based framework for fetal cortical surface reconstruction. **SuD-CoTAN** builds on CoTAN framework (Ma et al., 2023), which produces topologically consistent surfaces. Unlike CoTAN, **SuD-CoTAN** also provides vertex-wise anatomical consistency, where reconstructed surfaces are directly aligned to weekly fetal templates. Additionally, **SuD-CoTAN** is robust to variations in acquisition protocol, producing consistent surface measures across different fetal datasets. We evaluate our approach on fetal MRI scans from both normative samples from the developing Human Connectom Project (dHCP) and controls from the Intelligent Fetal Imaging and Diagnosis (iFIND) clinical research study, demonstrating geometric accuracy and cross-dataset harmonisation. Importantly, the iFIND dataset is entirely unseen during training, and it is acquired on a different scanner, with a different protocol. As such, performance on iFIND directly reflects out-of-distribution generalisation. Results show the method achieves superior surface precision and enhanced harmonisation relative to baseline methods, with sufficient coherence across acquisitions to support the high-resolution (per-vertex) normative modeling of sulcal depth.

## 2. Methods

Our proposed single-step surface reconstruction and template alignment framework is illustrated in Figure 2. While traditional pipelines depend on brain tissue segmentations to extract cortical surfaces, and rely on spherical registration to templates to enable vertex-wise comparisons, our method operates on fetal MRI directly, and integrates template alignment

into the reconstruction, to produce analysis-ready, template-aligned surfaces from one processing step. This pipeline is described in Figure 3, with an architecture that builds from CoTAN (Ma et al., 2023) but introduces key methodological and data processing advances that: 1) introduce a novel **SulcNet** module that estimates sulcal depth during training to guide and constrain template alignment; 2) adapt training with augmentations to achieve harmonisation; and 3) allow CoTAN to be run successfully for fetal data for the first time.

### 2.1. Generation of pseudo-ground truth surfaces and initial templates

CoTAN (Ma et al., 2023) relies on the availability of pseudo-ground truth (pGT) surfaces to supervise training, as well as an initial cortical template used to initialise surface reconstruction. However, pGT surfaces are non-trivial to obtain (Ma et al., 2024), due to the lower image quality of fetal MRI stemming from challenges associated with scanning free-moving fetuses within the maternal body, where volumetric data need to be reconstructed from stacks of acquired 2D slices (Wright et al., 2018; Cordero-Grande et al., 2019; Kuklisova-Murgasova et al., 2012; Uus et al., 2025). To address these challenges, we employ a fetal-specific processing workflow consisting of the following steps. First, $T_2$-weighted ($T_2$w) fetal MRI volumes are segmented using BOUNTI (Uus et al., 2023), an automated brain tissue parcellation model for 3D T2w fetal MRI, adapted from our neonatal protocols (Makropoulos et al., 2014, 2016) through extensive manual correction. Second, using the BOUNTI-derived tissue labels, approximate pGT white and pial cortical surfaces are generated by adapting our neonatal surface extraction pipeline (Schuh et al., 2017) to fetal anatomy.

Fetal cortical templates were generated separately for each hemisphere and for each week of gestation (from 25–36 weeks) using Multimodal Surface Matching (MSM) (Robinson et al., 2014, 2018; Besenczi et al., 2024; da Silva et al., 2025). Template construction follows established fetal atlas methodologies (Kuklisova-Murgasova et al., 2011; Serag et al., 2012; Bozek et al., 2018; Williams et al., 2023; Karolis et al., 2023) and consists of iterative surface alignment and kernel-weighted averaging with temporally adaptive regression, yielding one template per gestational week, which is subsequently aligned to the 36-week space through template-to-template alignment. Each fetal template comprises inner (white) and outer (pial) cortical surfaces, inflated surfaces, spherical representations, and associated cortical feature maps, including sulcal depth, curvature, and cortical thickness. For the purpose of this study, however, we utilise the inflated surfaces and their corresponding sulcal depth maps as initial weekly templates, replacing the single initial template mesh used in the original CoTAN framework (Ma et al., 2023).

### 2.2. SuD-CoTAN: Sulcal Depth–guided Cortical Surface Reconstruction

Our proposed **SuD-CoTAN** takes as input a fetal $T_2$w MRI scan and the gestational age (GA) of the subject, and predicts a conditional time-varying velocity field (CTVF) that deforms the closest-in-age initial surface into the subject-specific white matter (WM) (inner) surface (see Figure 3). To enable single-step surface fitting while implicitly enforcing surface-template correspondence, **SuD-CoTAN** integrates **SulcNet** as a feature-based guidance module. Specifically, through minimizing the normalised cross correlation (NCC) between the predicted sulcal depth map and an age-matched template sulcal depth map, **SulcNet**

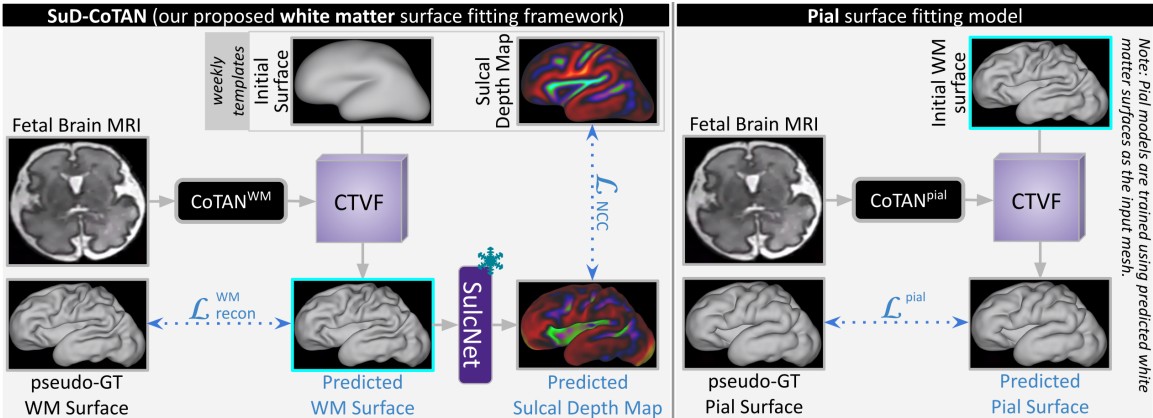

Figure 3: Overview of the **SuD-CoTAN** framework. Left: The model takes fetal brain MRI as input and deforms a gestational-age–specific template surface into a subject-specific white matter (WM) prediction. A pre-trained **SulcNet** provides sulcal-depth guidance to ensure consistent deformations across subjects. Right: For pial surface reconstruction, the initial meshes are subject-specific predicted WM surfaces and are subsequently deformed into pial surfaces as proposed in (Ma et al., 2024).

encourages anatomically consistent deformations. For pial (outer) surface reconstruction, a second CoTAN-based network is trained following the formulation introduced in (Ma et al., 2024). In this stage, the initial surface is given by the predicted WM surface from the pre-trained **SuD-CoTAN** model, which is then further deformed via a learned CTVF to match the pGT pial surfaces.

### 2.3. SulcNet: Sulcal Depth Prediction Network

**SulcNet** itself is a spherical U-Net (Monti et al., 2017; Zhao et al., 2019), adapted from (Suliman et al., 2022), that takes the 3D coordinates of the deformed WM surface as input and outputs average surface convexity (or sulcal depth) at each vertex. The input to this network corresponds to vertex coordinates from white matter meshes that have been resampled to the resolution and mesh topology of a sixth-order icosphere (with 40,962 vertices) - taking advantage of implicit vertex correspondence of spheres and anatomical (WM) surfaces to define convolutional kernels on the sphere, while learning from coordinates that reflect the native WM geometry. The architecture then follows a six-stage encoder–decoder design, with each resolution defined by progressively coarser/finer icospheres (Suliman et al., 2025), learning MoNet convolutions (Monti et al., 2017) with LeakyReLU activations ($\alpha = 0.2$), hexagonal mean-pooling and transpose-convolutions for upsampling (Zhao et al., 2019). A compact feedforward layer (Suliman et al., 2022) produces the final sulcal depth prediction per vertex. These generated sulcal depth maps are then used to guide template matching through integration of a sulcal depth loss (see Section 2.4).

## 2.4. Optimisation

**SuD-CoTAN** is trained with a combination of four losses (Ma et al., 2023, 2024): a bidirectional Chamfer (bi-Chamfer) distance loss ($\mathcal{L}_{cd}$) (Bongratz et al., 2022) between the predicted and pGT vertices, and three smoothness penalties: a normal consistency loss $\mathcal{L}_{nc}$, constraining cosine similarity between adjacent face normals, the mesh Laplacian loss $\mathcal{L}_{lap}$ to promote smoothness, and an edge length loss $\mathcal{L}_{edge}$ to discourage irregular or excessively stretched edges. During training, the pre-trained **SulcNet** outputs a sulcal depth map ($SD_{predicted}$) from each predicted WM surface, which is then compared with that of the closest-in-age template ($SD_{template}$) using NCC. The total WM loss becomes:

$$\mathcal{L}^{WM} = \mathcal{L}_{cd} + \lambda_{lap}\mathcal{L}_{lap} + \lambda_{nc}\mathcal{L}_{nc} + \lambda_{edge}\mathcal{L}_{edge} + \lambda_{SD}\mathcal{L}_{NCC}(SD_{predicted}, SD_{template}) \quad (1)$$

For pial surfaces, we follow the previously proposed method from (Ma et al., 2024), where the Chamfer distance is replaced with a single-directional Chamfer loss $\mathcal{L}_{cd-1d}$ between predicted and pGT pial surfaces. Two additional penalties are applied: an inflation loss $\mathcal{L}_{infl}$, which constrains surface inflation to follow normal directions (Ma et al., 2024), and a pial-outside-WM loss $\mathcal{L}_{pow}$ (Appendix B), which penalizes pial vertices whose displacement vectors point inward relative to WM normals. The total pial loss becomes:

$$\mathcal{L}^{pial} = \mathcal{L}_{cd-1d} + \lambda_{lap}\mathcal{L}_{lap} + \lambda_{nc}\mathcal{L}_{nc} + \lambda_{edge}\mathcal{L}_{edge} + \lambda_{infl}\mathcal{L}_{infl} + \lambda_{pow}\mathcal{L}_{pow} \quad (2)$$

**SulcNet** uses a Smooth L1 loss and it is trained independently on pGT meshes and sulcal depth maps derived from normative samples using our classical surface processing pipeline (Uus et al., 2023; Schuh et al., 2017; Makropoulos et al., 2018; Schuh et al., 2017). The trained **SulcNet** is then frozen during optimisation of the **SuD-CoTAN** network. The final model was selected based on the lowest validation mean squared error (MSE) and NCC.

**Training.** WM and pial models were trained for 400 epochs using the Adam optimizer. Following (Ma et al., 2023, 2024), we first pre-train the models using relatively large loss weights, then fine-tune them with smaller weights (Appendix C and Appendix G for choice of $\lambda_{SD}$). During training, we apply MONAI (Cardoso et al., 2022) augmentations designed to mimic realistic variations encountered in fetal MRI, namely, bias-field inhomogeneities, Gaussian noise, Gaussian smoothing, gamma contrast adjustments, and random histogram shifts (see Appendix D for more details on how these were parameterised). **SulcNet** was trained for 1000 epochs using an Adam optimizer, incorporating geometric augmentations consisting of small random translations and rotations of the meshes (see Appendix D for more details).

**Evaluation.** We evaluate our proposed framework in four ways: first, we assess **SulcNet** for its suitability for anatomical guidance during **SuD-CoTAN** training (see Section 3.2); second, we analyse population-level consistency and developmental trajectories across our cohorts (see Section 3.3); and third, we evaluate cross-dataset harmonisation using global surface-based cortical metrics (see Section 3.4). Finally, we evaluate **SuD-CoTAN** against three related configurations (see Table 1) to isolate the effects of data augmentation (Section 3.5). In summary: (i) the baseline CoTAN model uses inflated weekly templates as initial surfaces and no augmentation; (ii) CoTAN + augmentation

Table 1: Ablation study model configurations for WM and pial surface reconstruction.

| Model | Augmentation | Sulcal Depth Guidance |
|---|:---:|:---:|
| CoTAN (baseline) | – | – |
| CoTAN+aug | ✓ | – |
| CoTAN+SD | – | ✓ |
| **SuD-CoTAN (proposed)** | ✓ | ✓ |

(CoTAN+aug) incorporates intensity and contrast augmentations; (iii) the CoTAN + sulcal depth guidance (CoTAN+SD) adds **SulcNet** guidance without augmentation; and (iv) our proposed **SuD-CoTAN** combines augmentations with **SulcNet** guidance.

## 3. Experiments and Results

### 3.1. Fetal MRI Data Selection and Preprocessing

We evaluate our framework on two fetal $T_2$w MRI datasets, using one for training (dHCP) and the other exclusively for testing (iFIND[1]). All scans were motion corrected and reconstructed using slice-to-volume registration (SVR) to 0.5mm isotropic resolution (Kuklisova-Murgasova et al., 2012; Wright et al., 2018; Cordero-Grande et al., 2019; Uus et al., 2025). For all experiments, $T_2$w images were affinely aligned to a 36-week fetal brain atlas (Uus et al., 2021), and in this study we only consider the left hemisphere.

**Training Dataset.** The training and validation dataset consists of 210 dHCP scans (20.86 - 38.29 weeks GA) acquired on a 3T Philips Achieva MRI system with a 32-channel cardiac coil ($T_E = 250$ms, $1.1 \times 1.1$mm in-plane resolution, and 2.2mm slice thickness) using a dedicated protocol (Price et al., 2019). We use 200 subjects for training and 10 for validation (see Table A.1, "Training" rows). Both our cortical surface reconstruction models and the sulcal depth prediction network were trained exclusively on these dHCP training subjects.

**Testing Dataset.** Additional 20 dHCP subjects were reserved for testing, together with 173 iFIND subjects (22.29 - 31.86 weeks GA) scanned on a 1.5T Philips Ingenia MRI system using a 28-channel torso coil ($T_E = 80$ms, $1.25 \times 1.25$mm in-plane resolution, and 2.5mm slice thickness). We select 20 iFIND subjects, age-matched to the dHCP test set to avoid confounding generalisation performance with gestational age effects, and generate pseudo-ground truth surfaces using the same fetal-specific pipeline, consisting of BOUNTI-based tissue segmentations, followed by cortical surface reconstruction with a modified neonatal dHCP pipeline (see Section 2.1). We use these dHCP and iFIND subjects to evaluate the performance of our proposed model (Table 2), as well as for testing **SulcNet** generalisation (Figure 4). For more details on data partition see Table A.1, "Evaluation" rows. For subsequent cortical metric analyses, we select a subset of 140 subjects from the entire dHCP dataset to match the GAs of the 173 iFIND subjects (see Table A.1, "Feature Extraction" rows), ensuring that age-related effects do not confound cross-dataset comparisons.

---

1. data released on NIMH Data Archive (NDA), collection 5690: nda.nih.gov/edit_collection.html?id=5690

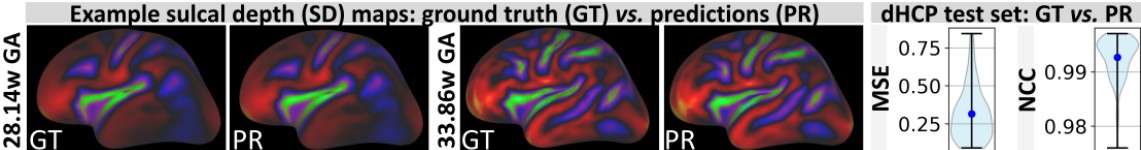

Figure 4: Predictions from **SulcNet** showing (left): two examples comparing ground truth (GT) *vs.* predicted (PR) sulcal depth (SD) maps; and *right*: quantitative evaluation of mean squared error (MSE) and normalised cross correlation (NCC) between GT and PR for dHCP test examples. Blue dots represent the mean.

### 3.2. SulcNet: Evaluation of Predicted Sulcal Depth Maps

First, we evaluate **SulcNet** to quantify its accuracy in sulcal depth prediction and thus suitability for guiding surface reconstruction. Figure 4 presents qualitative (left) and quantitative (right) results. Visually, the two representative examples (28.14 and 33.86 weeks GA) show strong agreement between ground-truth and predicted sulcal depth maps. Moreover, **SulcNet** achieved a low MSE of $0.32 \pm 0.17$mm and a high NCC of $0.99 \pm 0.01$, indicating strong correspondence between predicted and pGT sulcal depth maps. The high NCC values are particularly important, as our proposed cortical surface reconstruction model relies on predicted sulcal depth maps that remain closely aligned with age-specific template sulcal depth patterns in terms of NCC. These results suggest that **SulcNet** provides the consistency and accuracy required for reliable, age-aware guidance during cortical surface reconstruction.

### 3.3. Population-level Consistency and Developmental Trajectories

To assess whether **SuD-CoTAN** produces more developmentally consistent population-level representations, we generated age-stratified average and standard deviation sulcal depth maps from the unseen iFIND test dataset subjects (Table A.1, "Feature Extraction" rows) across gestational ages 22–32 weeks (Figure 5). Figure 5A presents example average and standard deviation sulcal depth maps projected onto inflated surfaces. Visually, mean sulcal depth maps are better defined with lower standard deviation when sulcal depth guidance is used (**SuD-CoTAN**). Figure 5B plots the distribution of variance (left) and the mean sharpness (right) across the surface, for all GAs. These results show that **SuD-CoTAN** yields higher sharpness and lower variability relative to the CoTAN+aug model. For similar results on curvature, see Appendix E, while a combined dHCP+iFIND analysis can be found in Appendix H and Appendix I.

To further quantify developmental coherence, we modeled sulcal depth trajectories across the 313 dHCP and iFIND subjects (Table A.1 "Feature Extraction") using vertex-wise Gaussian Process (GP) regression, fitted using the GPyTorch variational framework (Gardner et al., 2018), using a combination of linear and Radial Basis Function (RBF) kernels. We compared sulcal depth outputs from the CoTAN+aug and **SuD-CoTAN** models, and assessed accuracy using the mean absolute error (MAE) between GP-predicted and the subject-specific sulcal depth values calculated from the reconstructed cortical surfaces.

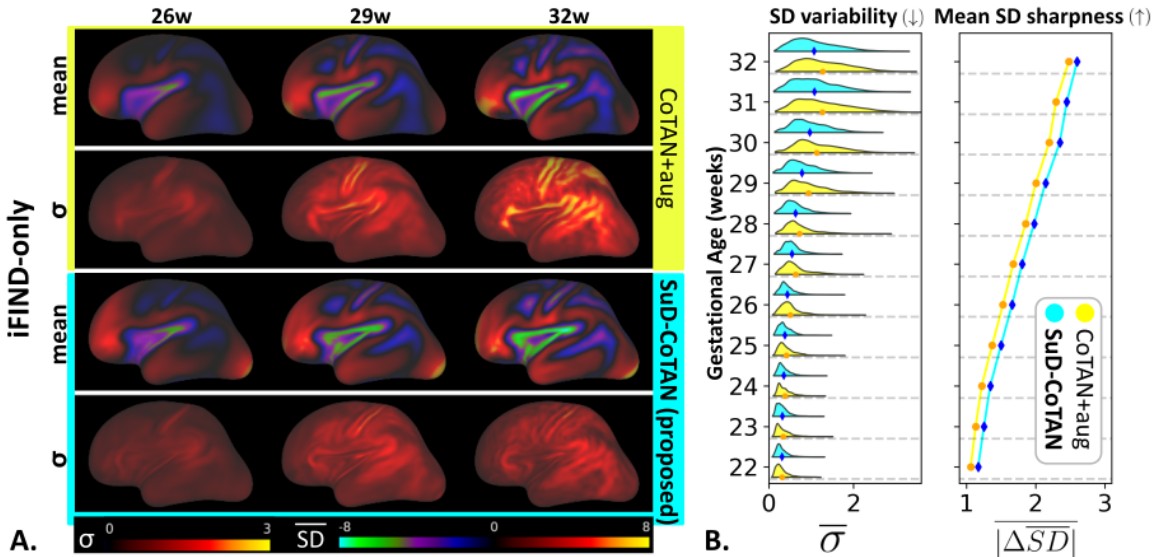

Figure 5: Sulcal depth maps across gestational ages for the iFIND dataset. **A.** Example mean ($\overline{SD}$) and standard deviation ($\sigma$) sulcal depth (SD) maps for subjects scanned at 26, 29 and 32 weeks for the CoTAN+aug model (first two rows) and our proposed **SuD-CoTAN** model (last two rows). **B.** Sulcal depth maps variability (lower is better) and the mean absolute Laplacian of the average sulcal depth map (as a proxy for sharpness, where higher is better) for each gestational age, comparing the CoTAN+aug (yellow) and our proposed model (cyan).

Figure 6 showcases representative trajectories at two anatomical locations (gyral crown in orange and sulcal fundus in blue), alongside whole-brain GP means and MAE maps. Relative to CoTAN+aug, **SuD-CoTAN** yields lower MAE and trajectories with clearer separation between gyral and sulcal developmental trends, representing a deepening of the sulcal fundus and a rising gyral crown with GA. These patterns indicate that the proposed model better captures anatomically meaningful developmental trajectories at the population level.

### 3.4. Harmonisation of Surface-based Metrics

To evaluate cross-dataset harmonisation, we analysed global cortical metrics from the predicted surfaces of the same 313 dHCP and iFIND subjects, including mean cortical thickness, mean absolute sulcal depth, mean absolute curvature, and average surface area. Figure 7 shows the average metrics as a function of GA corresponding to the proposed **SuD-CoTAN** model (see Appendix F for the other three models). Polynomial regression controlling for GA revealed no significant cohort effects in terms of cortical thickness, sulcal depth, and surface area, while mean curvature still shows a minor cohort difference. As opposed to the CoTAN and CoTAN+SD models, **SuD-CoTAN** showed the strongest improvement in harmonising cortical thickness, an encouraging result given its reliance on accurate white and pial surface reconstructions. Notably, despite using sulcal-depth guidance, the

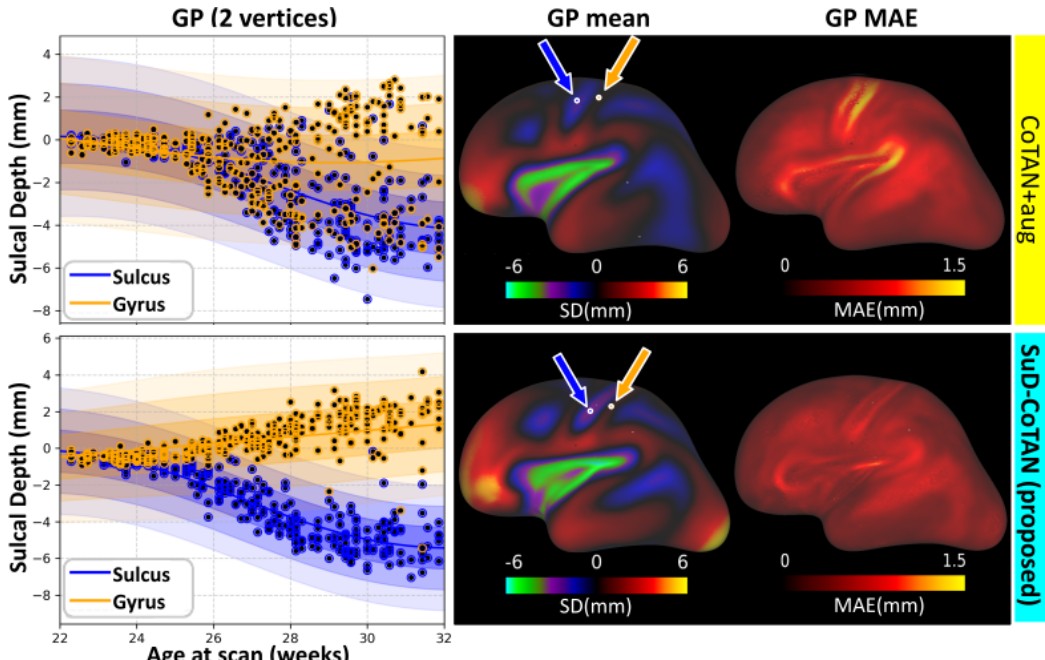

Figure 6: Gaussian Process (GP)-derived sulcal depth (SD) trajectories. Representative developmental trajectories at a gyral crown (orange) and a sulcal fundus (blue), alongside GP mean maps and mean absolute error (MAE) maps for the CoTAN+aug and **SuD-CoTAN** models.

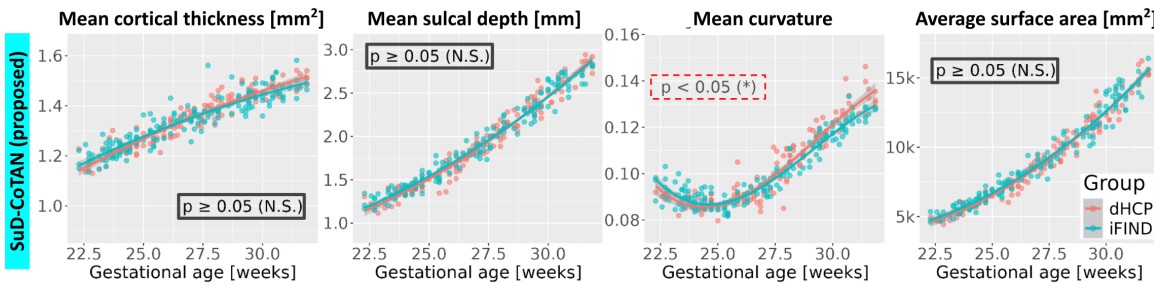

Figure 7: Global average cortical metrics as a function of gestational age. Dashed red boxes indicate metrics showing a statistically significant cohort effect between dHCP (red) and iFIND (blue) predicted surfaces, while black boxes mark metrics where this cohort effect is non-significant.

CoTAN+SD model did not harmonise mean sulcal depth across datasets, highlighting the importance of augmentation.

Table 2: White matter (WM) and pial surface predictions on the dHCP and iFIND test datasets, showing average symmetric surface distances (ASSD) and 90th percentile of Hausdorff distances (HD) between the predicted and the pseudo-ground truth (pGT) surfaces, and the ratio of self-intersecting faces (SIFs) in the predicted surfaces. First row shows average standard deviation across mean sulcal depth maps ($\overline{\sigma(\text{SD})}$) calculated from the predicted white matter surfaces. Values in bold represent best performing models ($p<.05$).

| | **Metric** | CoTAN (baseline) | CoTAN+SD | CoTAN+aug | SuD-CoTAN (proposed) |
|---|---|---|---|---|---|
| **SD** | $\overline{\sigma(\text{SD})}$ ↓ | 1.16 | **1.13** | 1.20 | **1.14** |
| **WM** | ASSD ↓ | 0.16±0.10 | 0.18±0.10 | **0.14±0.10** | **0.15±0.10** |
| | HD ↓ | 0.40±0.30 | 0.44±0.30 | **0.29±0.10** | **0.30±0.10** |
| | SIF(%) ↓ | **0.00±0.00** | 0.01±0.01 | 0.01±0.03 | 0.0003±0.001 |
| **Pial** | ASSD ↓ | 0.15±0.10 | 0.17±0.10 | **0.14±0.01** | **0.14±0.01** |
| | HD ↓ | 0.36±0.14 | 0.41±0.19 | **0.30±0.07** | **0.30±0.07** |
| | SIF(%) ↓ | **0.001±0.001** | 0.03±0.09 | 0.03±0.09 | 0.01±0.02 |

### 3.5. Ablation Study

The results of our ablation study are summarised in Table 2, where we assess geometric accuracy (average symmetric surface distances (ASSD) and 90th percentile of Hausdorff distances (HD)), mesh quality (self-intersecting faces (SIF)), and the stability of predicted sulcal depth maps in the template space, quantified as the average standard deviation across mean sulcal depth (SD) maps computed from the predicted WM surfaces ($\overline{\sigma(\text{SD})}$). All results were calculated on combined dHCP and iFIND test sets.

We observe that sulcal-depth guided models (**SuD-CoTAN** and CoTAN+SD models) achieved the lowest sulcal-depth variability, indicating that **SulcNet** guidance improves anatomical consistency without compromising surface integrity. Removing augmentation significantly increases geometric errors due to poor generalisability to iFIND dataset (see iFIND example in Figure 1), highlighting its importance for cross-dataset generalisation. Mesh quality remained high across all settings, with SIF equal to zero for the baseline CoTAN model, nearly zero for **SuD-CoTAN**, and only slightly higher in the other models. Although CoTAN+aug and **SuD-CoTAN** exhibit similar geometric accuracy, our primary goal was not to further optimise surface fitting error, but to ensure that introducing sulcal depth guidance as an additional anatomical constraint does not degrade surface reconstruction quality. In fact, the added SulcNet-based supervision preserves surface fidelity while enabling anatomically informed, template-aligned representations suitable for vertex-wise population analysis.

## 4. Discussion and Conclusions

This study proposed **SuD-CoTAN**, a sulcal-depth–guided framework for fetal cortical surface reconstruction, delivering anatomically informed surfaces with improved geometric

accuracy, sharper population-level maps, and more consistent developmental trajectories. Harmonisation analysis further showed that the method reduces cohort differences in cortical metrics, producing more comparable measurements across dHCP and iFIND. Moreover, GP modelling of sulcal depth trajectories reveals clearer developmental patterns, with deepening sulci and rising gyral crowns over gestation, supporting the biological plausibility of the reconstructed surfaces. These results indicate that anatomically informed guidance provides stable and biologically meaningful constraints that improve cross-dataset generalisation without compromising surface quality.

To achieve these results, our proposed **SuD-CoTAN** incorporates three key contributions. First, it combines surface reconstruction with sulcal depth-driven alignment across individuals, ensuring that reconstructed surfaces are template-aligned and vertex-wise comparable. This allows high-resolution screening for cortical alterations in development, enabling each individual's cortex to be compared directly against a normative cohort in a biologically interpretable way. Second, the method employs augmentation-based harmonisation across datasets, producing surfaces that are robust to variations in acquisition protocols, and ensuring consistent cortical metrics across both dHCP and iFIND. Third, **SuD-CoTAN** provides a unified, single-step framework that integrates reconstruction and alignment, avoiding multi-stage registration pipelines (e.g., MSM) that require additional software, multiple processing steps, and carry risks of cumulative errors. Collectively, these contributions allow clinicians and researchers to efficiently generate harmonised, anatomically aligned cortical surfaces ready for population-level analysis or clinical screening.

Despite these benefits, a key limitation of this work stems from the substantial natural variability in cortical folding. The human cortex exhibits diverse folding patterns, including branching, splitting, and tertiary folds, that cannot be fully captured by registration to a single population-average template (Guo et al., 2025). Future work will explore strategies that better accommodate individual folding diversity (Guo et al., 2025), as well as more powerful augmentation approaches, such as those proposed in FetalSynthSeg (Zalevskyi et al., 2024), to enhance robustness across datasets with significant domain shifts. Overall, our proposed model represents a promising step toward robust, harmonised fetal cortical surface analysis suitable for multi-site studies and future large-scale developmental neuroimaging applications.

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

## Appendix A. Dataset Split

Table A.1: Number of scans used for training and evaluating our models, as well as cortical feature extraction.

| Task | Dataset | Data Split | Subjects | GA (weeks) |
|---|---|---|---|---|
| **Training** | **dHCP** | **Train** | 200 | 29.1 ($\pm$3.88) |
| | **dHCP** | **Valid** | 10 | 28.9 ($\pm$6.03) |
| **Evaluation** | **dHCP** | **Test** | 20 | 28.5 ($\pm$3.79) |
| | **iFIND** | **Test** | 20 | 27.9 ($\pm$3.14) |
| **Feature Extraction** | **dHCP** | **Train/Valid/Test** | 140 | 27.3 ($\pm$2.49) |
| | **iFIND** | **Test** | 173 | 26.7 ($\pm$2.74) |

## Appendix B. Pial-outside-WM Loss

Letting $v_i^{wm}$ and $v_i^{pial}$ denote corresponding WM and pial vertices, and $n_i^{wm}$ the unit WM normal, we define:

$$\mathcal{L}_{pow} = \frac{1}{N} \sum_{i=0}^{N-1} \text{ReLU} \left( -n_i^{wm} \cdot \left( \widehat{v_i^{pial} - v_i^{wm}} \right) \right),$$ (B.1)

where $\widehat{(\cdot)}$ denotes vector normalisation. The ReLU ensures that only inward-pointing displacements (*i.e.*, violations) contribute to the loss.

## Appendix C. Training loss weights used for SuD-CoTAN

Table C.1: Weights for training SuD-CoTAN.

| Surface Type | Loss Term | Pre-training | Fine-tuning |
|---|---|---|---|
| **WM** | $\lambda_{\text{lap}}$ | 0.5 | 0.1 |
| | $\lambda_{\text{nc}}$ | $5 \times 10^{-4}$ | $1 \times 10^{-4}$ |
| | $\lambda_{\text{edge}}$ | $5 \times 10^{-4}$ | $1 \times 10^{-4}$ |
| | $\lambda_{\text{SD}}$ | 10 | 1 |
| **Pial** | $\lambda_{\text{lap}}$ | 0.5 | 0.1 |
| | $\lambda_{\text{nc}}$ | $5 \times 10^{-4}$ | $1 \times 10^{-4}$ |
| | $\lambda_{\text{edge}}$ | $5 \times 10^{-4}$ | $1 \times 10^{-4}$ |
| | $\lambda_{\text{infl}}$ | 2.5 | 2.5 |
| | $\lambda_{\text{pow}}$ | 2.5 | 2.5 |

## Appendix D. Data augmentations during training

Table D.1: Data augmentations applied during model training.

| Model | Augmentation | Parameters |
|---|---|---|
| **SuD-CoTAN** | Bias-field augmentation | degree = 3; coeff. $\sim \mathcal{U}(-0.5, 0.5)$ |
| | Gaussian noise | $\sigma \sim \mathcal{U}(0.0, 0.02)$ |
| | Gaussian smoothing | $\sigma \sim \mathcal{U}(0.5, 1.5)$mm |
| | Gamma correction | $\gamma \sim \mathcal{U}(0.75, 1.25)$ |
| | Histogram shifts | 3–5 random control points |
| **SulcNet** | Translation | $t_x, t_y, t_z \sim \mathcal{U}(-3, 3)$mm |
| | Rotation | $r_x, r_y, r_z \sim \mathcal{U}(-15, 15)°$ |

## Appendix E. Average Curvature Maps for the iFIND dataset

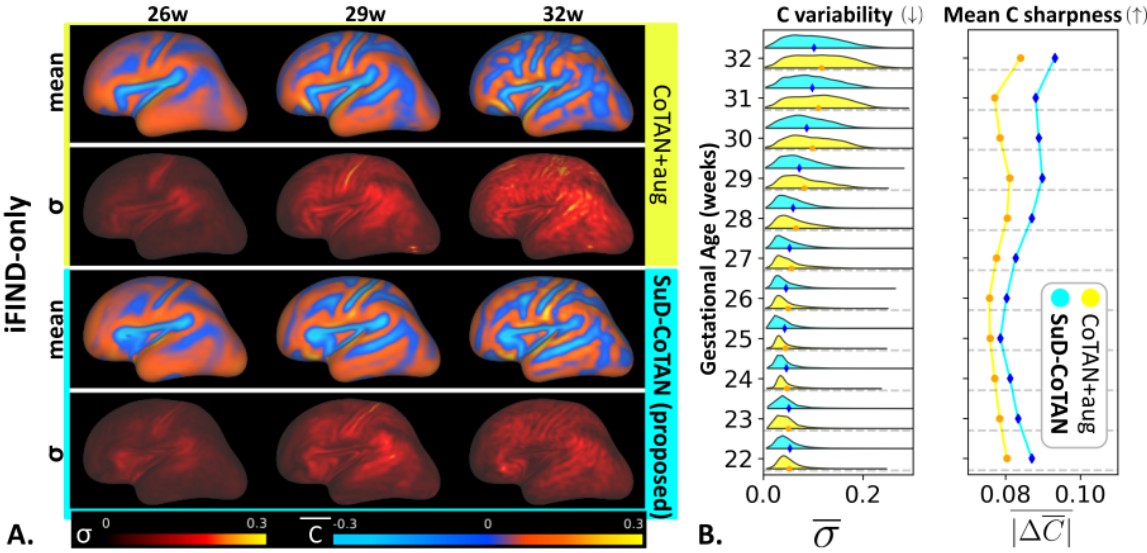

Figure E.1: Curvature maps across gestational ages for the iFIND dataset. **A.** Example mean ($\overline{C}$) and standard deviation ($\sigma$) curvature (C) maps for subjects scanned at 26, 29 and 32 weeks for the CoTAN+aug model (first two rows) and our proposed **SuD-CoTAN** model (last two rows). **B.** Curvature maps variability (lower is better) and the mean absolute Laplacian of the average curvature map (as a proxy for sharpness, where higher is better) for each gestational age, comparing the CoTAN+aug (yellow) and our proposed model (cyan).

# Appendix F. Harmonisation of Surface-based Metrics

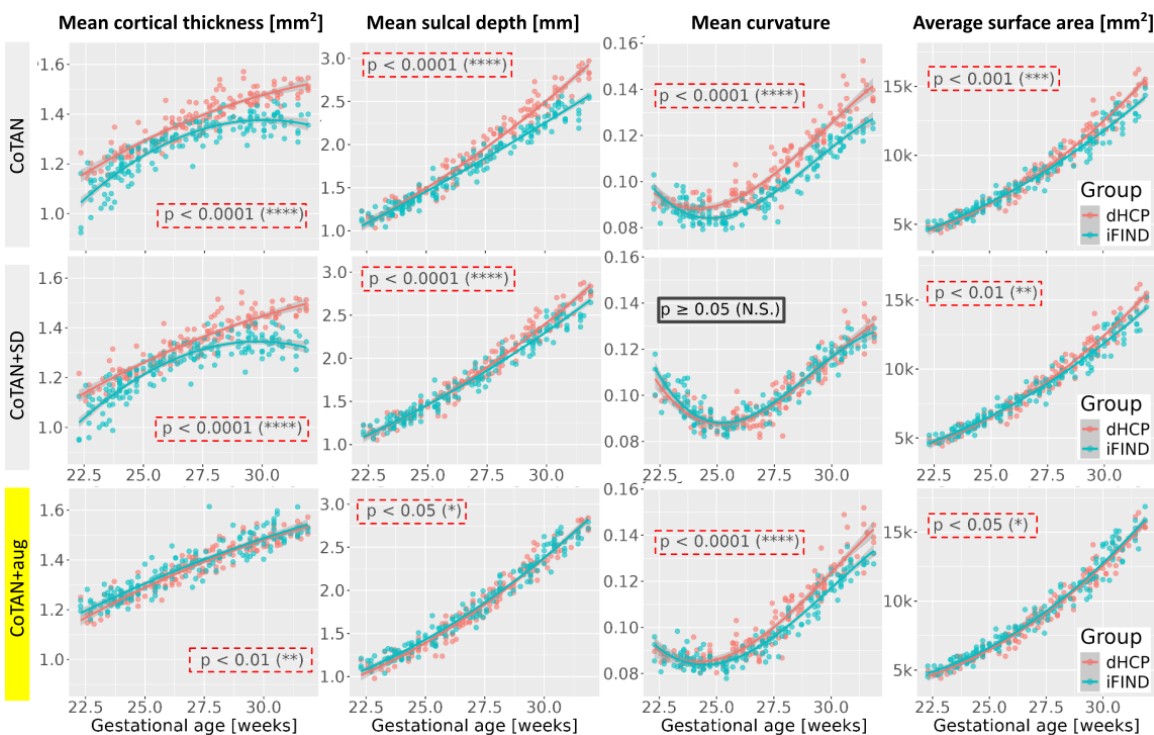

Figure F.1: Global average cortical metrics as a function of gestational age. Dashed red boxes indicate metrics showing a statistically significant cohort effect between dHCP (red) and iFIND (blue) predicted surfaces.

## Appendix G. Weighting for Sulcal Depth Guidance

An ablation study was conducted on dHCP data (training/validation/testing) using a reduced training schedule ($N_{epochs} = 100$) to select an appropriate value for $\lambda_{SD}$ when using **SulcNet** guidance. Table G.1 reports the mean ± standard deviation HD and ASSD between predicted and pGT surfaces, as well as mean ± standard deviation NCC scores between sulcal depth maps corresponding to the predicted surfaces and their closest-in-age template sulcal depth maps. Four configurations were evaluated: $\lambda_{SD} = 0$ (no guidance), $\lambda_{SD} = 1$ (constant), $\lambda_{SD} = 10$ (constant), and a two-stage schedule: $\lambda_{SD} = 10$ for the first 50 epochs, then $\lambda_{SD} = 1$ for the last 50 epochs. Overall, the two-stage strategy ($\lambda_{SD} = 10 \rightarrow 1$) provides an effective compromise by achieving strong anatomical alignment with the template while preserving high geometric fidelity to the pseudo-ground-truth surfaces.

Table G.1: Ablation study for selecting $\lambda_{SD}$ when training with **SulcNet** guidance. Bold values indicate best performance; italicized values indicate worst performance.

| $\lambda_{SD}$ | HD ($\downarrow$) | ASSD ($\downarrow$) | NCC ($\uparrow$) |
|---|---|---|---|
| 0 | **0.89 ± 0.19** | **0.43 ± 0.08** | *0.24 ± 0.05* |
| 1 | **0.92 ± 0.20** | **0.45 ± 0.08** | 0.46 ± 0.08 |
| 10 | *1.15 ± 0.24* | *0.55 ± 0.10* | **0.71 ± 0.05** |
| 10 → 1 | **0.86 ± 0.20** | **0.42 ± 0.08** | 0.69 ± 0.05 |

## Appendix H. Average Sulcal Depth Maps

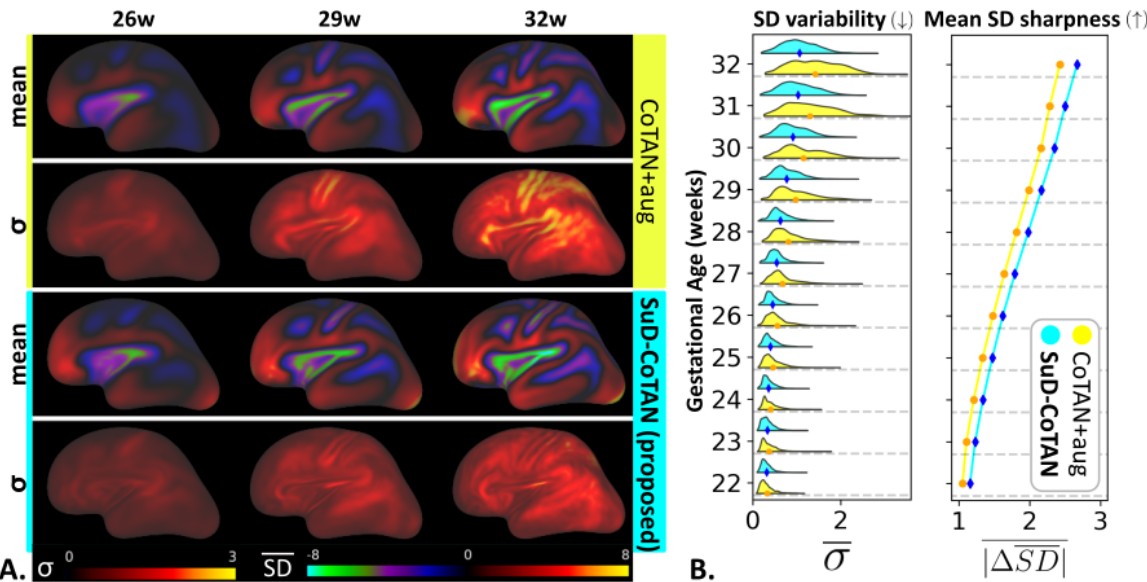

Figure H.1: Sulcal depth maps across gestational ages for subjects pooled from both dHCP and iFIND datasets. **A.** Example mean ($\overline{\text{SD}}$) and standard deviation ($\sigma$) sulcal depth (SD) maps for subjects scanned at 26, 29 and 32 weeks for the CoTAN+aug model (first two rows) and our proposed **SuD-CoTAN** model (last two rows). **B.** Sulcal depth maps variability (lower is better) and the mean absolute Laplacian of the average sulcal depth map (as a proxy for sharpness, where higher is better) for each gestational age, comparing the CoTAN+aug (yellow) and our proposed model (cyan) for each cohort individually.

## Appendix I. Average Curvature Maps

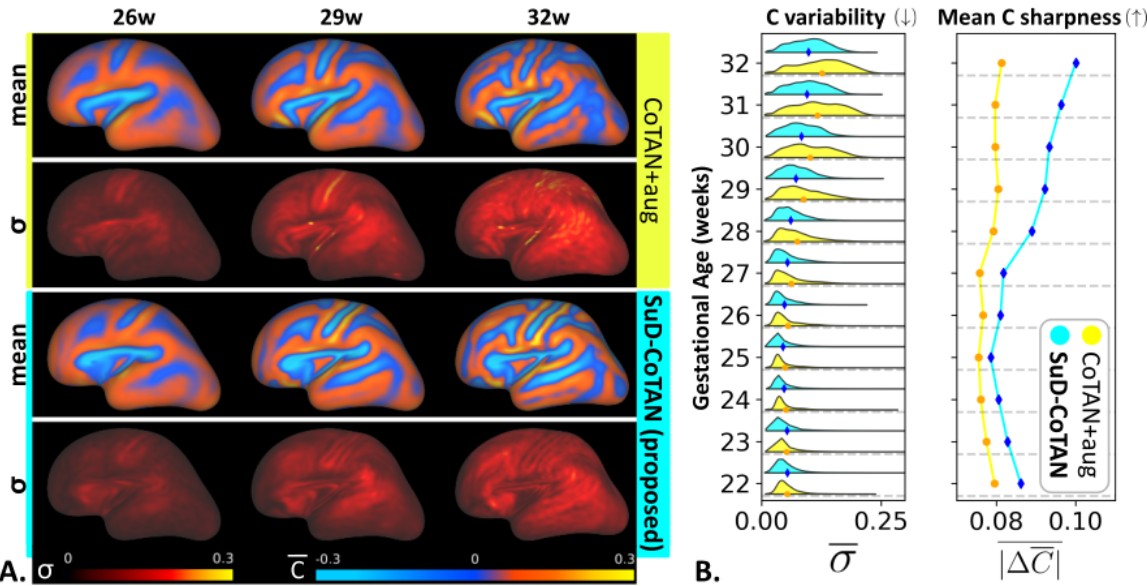

Figure I.1: Curvature maps across gestational ages for subjects pooled from both dHCP and iFIND datasets. **A.** Example mean ($\overline{C}$) and standard deviation ($\sigma$) curvature (C) maps for subjects scanned at 26, 29 and 32 weeks for the CoTAN+aug model (first two rows) and our proposed **SuD-CoTAN** model (last two rows). **B.** Curvature maps variability (lower is better) and the mean absolute Laplacian of the average curvature map (as a proxy for sharpness, where higher is better) for each gestational age, comparing the CoTAN+aug (yellow) and our proposed model (cyan) for each cohort individually.

