# OpenReview forum: "SuD-CoTAN: Sulcal Depth-guided Anatomically Consistent Fetal Cortical Surface Reconstruction"
_MIDL.io/2026/Conference — MIDL 2026 Poster_

### Official Review · Reviewer_gn6M · 2026-01-05

**Confidence:** 4
**Preliminary Rating:** 5
**Final Rating:** 5

**Summary:**

The paper introduces SuD-CoTAN, a deep learning framework that performs anatomically consistent fetal cortical surface reconstruction and template alignment in a single step using sulcal depth guidance. The paper introduces SuD-CoTAN, a deep learning framework that performs anatomically consistent fetal cortical surface reconstruction and template alignment in a single step using sulcal depth guidance.

**Strengths:**

1. The entire process requires only 3 seconds, which is remarkably fast for cortical reconstruction, thereby demonstrating significant potential for clinical applications.
2. The introduction of the SulcNet module provides explicit sulcal depth guidance, ensuring that the reconstructed surfaces maintain accurate vertex-wise anatomical correspondence, which is superior to methods that rely solely on geometric fitting.
3. The clinical utility of the proposed method is validated through experiments on population-level consistency and developmental trajectories.

**Weaknesses:**

1. Section 2.1, which describes the extraction of pseudo-ground truth surfaces and initial templates, is interspersed with numerous citations, making it somewhat convoluted and difficult to follow. It would be beneficial to summarize the procedure into discrete steps, for example: Step 1: xxx, Step 2: xxx.

2. It is unclear whether the iFIND dataset is publicly available. How were the ground truth cortical surfaces for iFIND obtained?

3. Why does the proposed method exclusively consider sulcal depth? Can other cortical metrics, such as curvature and thickness, not be directly derived?

4. Based on my experience, cortical surface reconstruction methods are typically sensitive to the hyperparameters of the loss function. Given that multiple loss functions are involved, how were the weights for training SuD-CoTAN determined in this study?

5. Why are different parameters employed for pre-training and fine-tuning?

**Detailed Comments:**

N/A

**Justification Of Final Rating:**

Thank you to the authors for their clarifications, which have resolved my concerns. Although other reviewers appear to consider the novelty of this work limited, I believe fetal brain MRI CSR is inherently challenging. Therefore, I maintain my current score and recommend acceptance. In addition, supervised method for fetal MRI may suffer from limited generalizability. For example, differences in image contrast between TSE and BTFE sequences, as well as variations introduced by different reconstruction methods (NeSVoR vs NiftyMIC), could affect performance. I encourage the authors to explore strategies to improve model generalization in future work. Finally, fetal brain CSR seems to lack a widely accepted gold-standard pipeline comparable to FreeSurfer for adult brain MRI.

**Justification Of The Preliminary Rating:**

Despite some limitations, given the scarcity of fetal cortical reconstruction tools and the fact that this study represents the first implementation of COTAN in fetal subjects with extensive validation through numerous downstream tasks, I recommend accepting this paper. Moving forward, I hope the open-source code tutorial can be made more detailed and that pre-trained weights will be provided.

**Questions To Address In The Rebuttal:**

See weaknesses

---

> ### Author Response · Authors · 2026-01-24
>
> We sincerely thank the reviewer for their thorough and thoughtful evaluation of our work, as well as for the positive assessment of its clinical relevance, computational efficiency, and anatomical consistency. Please note that in the revised manuscript the ablation study is now in Section 3.5. Below, we address each of the reviewer’s comments in detail.
>
> **Clarity of Section 2.1:** We completely agree with this comment regarding Section 2.1 which describes the generation of pseudo-ground truth surfaces and initial templates. We have revised the manuscript to summarise the workflow in steps which we hope has improved clarity.
>
> **The fetal iFIND MRI:** Thank you for raising this question, and as Reviewer 2 also pointed out, the original manuscript did not provide sufficient information about the iFIND dataset. The iFIND project is a UK-based initiative focused on improving fetal imaging and diagnosis, with standardized acquisition protocols and ethical approvals. More information can be found on the study website https://www.ifindproject.com/. The iFIND MRI data are available as part of a data release on the NIMH Data Archive (NDA), collection 5690: https://nda.nih.gov/edit_collection.html?id=5690. We have now added this extra information in the revised manuscript. In terms of the pseudo-GT surfaces, both iFIND and dHCP pGT were generated using the processing described in Section 2.1
>
> **Focus on Sulcal Depth vs. Other Cortical Metrics:** We decided to use sulcal depth because curvature becomes far too variable across individuals as gestation proceeds (Guo et al., 2025). Addressing the challenges of defining normative development in the face of such variation is a key focus of future work. However, it is possible to include different cortical features in our framework without any modifications to the architecture.
>
> **Loss Weight Selection and Pre-training vs. Fine-tuning Strategy:** We acknowledge that cortical surface reconstruction frameworks can be sensitive to the choice of loss weights and training schedules. In our study, the selection of loss weights and the transition from pre-training to fine-tuning follow the strategy used prior in CoTAN, as this staged optimization has been shown to improve training stability and surface quality.
>
> Regarding the weighting of the sulcal depth (SD) guidance term, we conducted a dedicated ablation study (now included in Appendix G: Weighting for Sulcal Depth Guidance, and referred to in Section 2.4 Optimisation) using a reduced training schedule of 100 epochs. This experiment evaluated different values of the SD loss weight, by assessing both geometric reconstruction accuracy (Hausdorff Distance, HD; Average Symmetric Surface Distance, ASSD) and template alignment (as normalised cross correlation, NCC, between subject and closest-in-age template sulcal depth maps).
>
> The results show that no guidance (i.e., lambda_SD = 0) yields strong geometric reconstruction (low HD/ASSD) but the poorest template alignment (low NCC), indicating weak anatomical correspondence. When kept constant, strong SD guidance (i.e., lambda_SD = 10) achieves excellent template alignment (high NCC) but obtains the worst surface reconstruction accuracy. Moderate constant SD guidance (i.e., lambda_SD = 1), achieves good reconstruction quality but provides only modest template alignment.
> The 2-stage weighting strategy (i.e., lambda_SD = 10 for the first half of training, followed by lambda_SD = 1 for the second half) preserves geometric accuracy (HD and ASSD comparable to lambda_SD = 0), while achieving near-optimal template alignment (NCC = 0.69 vs. 0.71 for constant lambda_SD = 10). We therefore considered this strategy to be an effective compromise for achieving both anatomical consistency and surface fidelity. We have clarified this and added the ablation results to the supplementary material (Appendix G).
>
> Bibliography:
> Guo et al., 2025 - https://www.biorxiv.org/content/biorxiv/early/2025/03/05/2025.03.04.641459.full.pdf

---

> > ### Comment · Area_Chair_Db6H · 2026-02-01
> > **Update the final ratings**
> >
> > Please kindly review the authors’ rebuttal and update the final rating by February 1, 2026 (23:59 AoE).

---

### Official Review · Reviewer_vM2b · 2026-01-10

**Confidence:** 5
**Preliminary Rating:** 3
**Final Rating:** 4

**Summary:**

The authors introduce sulcal depth-guided CoTAN (SuD-CoTAN), a framework that produces topologically correct and anatomically accurate WM and pial surfaces from fetal MRI images. It performs the surface reconstruction and alignment to underlying age-matched templates in one single step (as opposed to the multi-step approach that many SOTA tools take). The submission also introduces the SulcNet (feature-based guidance module) module that estimates sulcal depth during training to guide and constrain template alignment. The hope is that the new framework can facilitate the study of early neurodevelopmental trends without a need for any further processing (and spatial alignment steps). The submission only presents work on the left hemi, but that is not a theoretical constraint.

**Strengths:**

Developing robust and accurate tools on fetal brain images of high interest.

The proposed method could enable deriving meaningful developmental trajectories at the population level.

As the proposed framework reconstruct surfaces in spatial correspondence with age-matched templates, a vertex-wise anatomical correspondence is guaranteed in the population.

The experimental work demonstrates population-level consistency and clear development trajectories. Additionally, SuD-CoTAN showed the strongest improvement in harmonising surface-based metrics across data sets.

**Weaknesses:**

Moderate technical innovation is introduced by building on an existing DL solution (coTAN) with new guidance module. The authors themselves mention the limitations and also the reliance on the discreate atlases.

Very little is shared about one of the datasets (iFIND) used in the testing of the proposed framework. Is there a citation / website that can refer to this?

Some details of the experimental design are not clear. For example, the template definition or the data set that was used for 3.3's SulcNet evaluation? (is this using the 20 subset or the large data set?). For more questions resulting from lack of clarity see "Questions"

**Detailed Comments:**

The goal is to develop a robust surface reconstruction tool, generating anatomically and topologically correct meshes. The current framework builds on CoTAN and applied it to fetal MRI images with an additional sulcal depth estimating module.

The resulting surfaces are harmonized across acquisition protocols / datasets and automatically aligned to an age-matched template.

The framework requires weekly fetal templates that are constructed from the dHCP data based on previously published tools / pipeline. 12 pseudo ground truth (pGT) templates are generated. How is a "template" defined? Is it just a pial and wm surface or some associated properties as well? Will these be shared?

iFIND seems to encompass a smaller age range than dHCP. Were the dHCP test subjects selected with that info in mind? Given that the iFIND test subjects were age-matched to these, I assume that is the case.

re cortical metric analysis: I am a bit confused here. Are the authors using almost the full dHCP training data set for SuD-CoTAN for these experiments?  Why could not just use the iFIND data? Doesn't the current choice create a circular definition of surfaces and surface-based metrics?

How are the iFIND pGT surfaces? Same quality as the dHCP ones? Does that even matter for the performance of the proposed SuD-CoTAN?

Experiments: baseline+augmentaion performance is very close to that of the newly presented one (both on the pial and WM). The advantage of the new framework  seems to come from the resulting anatomical consistency.

Note: inconsistend --> inconsistent

**Justification Of Final Rating:**

I would like to thank the authors for their detailed rebuttal and providing additional information to support their submission.

I increased my rating to weak accept due to all the new details and clarification that went into the submission.

I still find the technical novelty low in the submission and the validation limited

**Justification Of The Preliminary Rating:**

Moderate technical contribution, but well executed data analysis on rare fetal MRI data.

Lower on technical novelty and various remaining questions regarding test data setup, data pre-processing and circularity of the study design.

**Questions To Address In The Rebuttal:**

12pGT: Are these good enough templates? Are you happy with them? Will these be shared? (Have they already been shared with dHCP-fetal?)

What is the template-to-template alignment choice for the pGT alignment to one central choice?

Are the pGT surfaces the same surfaces that were released with the dHCP_fetal cohort? If not, how are these different from those ones?

Do the authors have any comments /thoughts on how the proposed augmentation strategy compares to larger-scale, often unrealistic perturbations from the Synth literature?

Doesn't the decision of age-matching the dHCP and iFIND test sets prevent the authors from validating their tools on the full age range of dHCP?

Is 2.1 describing the same pipeline that was used before publicly releasing the dHCP fetal cohort with surface reconstructions?

How do you define "observed sulcal depth"?

---

> ### Author Response · Authors · 2026-01-24
>
> We sincerely thank the reviewer for their thoughtful and constructive evaluation of our work. We appreciate the positive assessment of the clinical relevance, population-level consistency, and harmonisation capabilities of the proposed SuD-CoTAN framework. As we believe that our original manuscript lacked clarity due to ordering of the experiments, please note that in the revised manuscript the ablation study is now in Section 3.5. We sincerely hope that we have addressed all of the reviewer’s questions and concerns.
>
> **The fetal iFIND MRI:** IFIND data are available as part of a data release on the NIMH Data Archive (NDA), collection 5690: https://nda.nih.gov/edit_collection.html?id=5690. We have now added this extra information in the revised manuscript.
>
> **Evaluation of SulcNet performance:** SulcNet was trained on the dHCP training dataset, and its performance was assessed on the test dataset (20 dHCP subjects + 20 iFIND subjects). We have now added this clarification in Section 3.1 of the revised manuscript.
>
> **Pseudo-GT surfaces used for training and evaluation:** The pseudo-GT surfaces used in our study are not the same as the cortical surfaces released with the dHCP_fetal cohort. In this study, the pGT surfaces were generated using a dedicated fetal-specific processing workflow which includes: 1) generating BOUNTI labels (Uus et al., 2023) from fetal T2w MRI data; and 2) the use of a modified version of the neonatal surface extraction pipeline (Schuh et al., 2017) to work with BOUNTI fetal labels. We have clarified this in Section 2.1 of the revised manuscript.
>
> **Average age-dependent surface templates:** Surface templates used in this paper are different from the ones available from gin-node https://doi.gin.g-node.org/10.12751/g-node.qj5hs7/, and comprise white and pial cortical surfaces, inflated surfaces and spheres, as well as cortical feature maps summarising sulcal depth, curvature and cortical thickness. For the purpose of this study, however, we only used the inflated surfaces and their corresponding sulcal depth maps as initial weekly templates, replacing the single initial template mesh used in the original CoTAN framework. We have clarified this in Section 2.1 of the revised manuscript.
>
> We recreated the templates using our latest in-house framework based on new BOUNTI segmentation and initial CoTAN training. The templates are good quality and generally consistent in time. However, we plan to further refine them based on our proposed SuD-CoTAN harmonised surface fitting, which will enable us to use a much larger database of subjects acquired using multiple acquisition protocols, and to make the new templates available online before the MIDL conference.
>
> **Augmentation strategy:** Our current augmentation strategy focuses on realistic variations commonly encountered in fetal MRI such as bias-field inhomogeneities, noise, smoothing and contrast adjustments. However, we view the Synth-style approach as a promising direction for future research, particularly for improving robustness to larger domain shifts and possibly pathological cases where data is much more scarce. We have revised the Discussion and Conclusions section to explicitly highlight this potential future research direction.
>
> **Age-matched test set:** Yes, the dHCP test subjects were selected with the age range of the iFIND cohort in mind, and the two test sets were explicitly age-matched. This was an intentional choice as the iFIND cohort does not extend beyond approximately 32 weeks gestation. Age-matching therefore ensures a fair and robust comparison. We have added this clarification in Section 3.1 (Testing Dataset.) of the revised manuscript.
>
> **Observed sulcal depth:** In our original manuscript, “observed sulcal depth” referred to the subject-specific sulcal depth values calculated from the reconstructed cortical surfaces. We understand that this was an ambiguous term and we have revised the manuscript (see Section 3.3) to clarify that these represent the subject-specific sulcal depth values.
>
> **Circularity of cortical feature comparison:** We acknowledge the reviewer’s concern about the potential circularity of using dHCP data both to train SuD-CoTAN and for the subsequent feature comparison analysis. For this reason, we now show intersubject consistency of cortical features (Figure 5) on the iFIND only in the revised manuscript (Section 3.3, Figure 5), and we moved the original figure which was combining subjects from both dHCP and iFIND datasets in the Appendix (see Appendix H for sulcal depth and Appendix I for Curvature). Please note that this new result shows similar anatomical consistency and trends with age as the combined analysis, supporting the robustness of our findings.
>
>
> Bibliography:
>
> Uus et al., 2023 - https://elifesciences.org/reviewed-preprints/88818v1
>
> Schuh et al., 2017 - https://ieeexplore.ieee.org/document/7950639

---

> > ### Comment · Area_Chair_Db6H · 2026-02-01
> > **Update the final ratings**
> >
> > Please kindly review the authors’ rebuttal and update the final rating by February 1, 2026 (23:59 AoE).

---

### Official Review · Reviewer_V7kh · 2026-01-16

**Confidence:** 4
**Preliminary Rating:** 3
**Final Rating:** 4

**Summary:**

This work presents a learning based method (SuD‑CoTAN) that directly fits anatomically consistent cortical meshes to T2‑weighted fetal MRI and aligns them to age‑matched templates in one step. Trained solely on normative dHCP data, it generalizes to new datasets and produces high quality fetal cortical surface reconstruction.

**Strengths:**

1) The problem motivation, existing methods, proposed method, datasets, experiments and implementation details have been presented clearly.

2) The manuscript presents a thorough and comprehensive evaluation of all aspects of the proposed method. The authors demonstrate the method’s performance across datasets and cortical features.

3) The baseline and its variants have been executed thoughtfully, providing a clear framework for understanding the incremental value of each variant.

**Weaknesses:**

1) The paper contains many abbreviations, and I strongly recommend that the authors spell out each term at its first occurrence including those used within the figures to improve clarity and readability.

2) I believe CoTAN is the true baseline, and I strongly recommend that the authors explicitly use the name “CoTAN” instead of the generic term “baseline” throughout the manuscript. For example, in Table 3, the first column should be labeled “CoTAN,” with “(Baseline)” placed beneath it for clarity. Similarly, for the last column, the header can be “(Proposed)” placed beneath SuD‑CoTAN!

3) From Table 1, CoTAN+Aug and SuD‑CoTAN show very similar performance, making it difficult to discern the added value of SulcNet within the proposed pipeline. This ambiguity is further reinforced by Fig. 5 and Fig. 6, where the observed improvements appear minimal. I encourage the authors to clarify the specific contribution of SulcNet both quantitatively and qualitatively and to explain how these seemingly modest gains translate into meaningful or clinically relevant improvements in practice.

**Detailed Comments:**

Please refer to summary, strengths and weaknesses.

**Justification Of Final Rating:**

Thanks to the authors for the detailed response to the raised comments. The revisions have addressed my concerns and have improved the readability of the manuscript. I am fine with increasing the score by one point.

**Justification Of The Preliminary Rating:**

The method is well‑designed and technically sound, but the improvements over the baselines are modest and not convincingly demonstrated. With the added value of SulcNet remaining unclear and the clinical relevance of the gains appearing limited, the overall contribution feels less substantial than expected.

**Questions To Address In The Rebuttal:**

I would strongly recommend authors to address the concerns raised in the weaknesses.

---

> ### Author Response · Authors · 2026-01-24
>
> We sincerely thank the reviewer for their thoughtful, constructive, and detailed evaluation of our work. We appreciate the positive assessment of our manuscript and we are grateful for the helpful suggestions aimed at improving the clarity, interpretability, and presentation of our contributions. Corresponding edits to the manuscript have been highlighted throughout. Below, we address each of the reviewer’s comments and recommendations.
>
> **Abbreviations:** We have carefully checked and corrected the manuscript to ensure abbreviations are spelled out at their first occurrence, as well as in all figures and table captions.
>
> **Naming of the compared techniques:** CoTAN is indeed the baseline and so have followed the reviewer’s suggestions and replaced “baseline” with CoTAN, “baseline+augmentation” to “CoTAN+aug”, and “baseline+SD” to “CoTAN+SD”, both in the manuscript, as well as in figure or table captions (Figures 1, 5, 6, 7; Table 1).
>
> **Clarification of technical contribution and performance improvement:** We would like to clarify that the purpose of introducing SulcNet and sulcal depth loss in our proposed framework (Sud-CoTAN) is to achieve consistent anatomical correspondences of the individual vertices, and not necessarily to improve quality of the surface fitting or harmonisation. While the original CoTAN accurately delineates cortical surfaces, the anatomical locations of the vertices are inconsistent between subjects as it fits a very smooth initial surface with no constraint on vertex locations. This is shown in our revised manuscript in Table 2 (first row) where sulcal-depth variability across the predicted surfaces is significantly higher in the original CoTAN and CoTAN+aug compared to CoTAN+SD and proposed SuD-CoTAN which include sulcal depth guidance. This is further visualised in Figure 6 where individual trajectories at the population level are much better captured by SuD-CoTAN then the original CoTAN. Such vertex-wise anatomical correspondences are essential for practical applications, because they allow comparison of cortical measures across individuals and anomaly detection without any further post-processing of fitted surfaces. Figure 5 visualises the differences in sharpness of the average templates (mean sulcal depth) and lower variability between subjects (standard deviation of the sulcal depth). While these differences might be perceived as subtle, they will have a significant impact on sensitivity of downstream analyses, as shown by poor separation of trajectories in Fig. 6 when CoTAN is used.
>
> We understand though why our contribution was not clear from the manuscript due to ordering of the experiments. To make our contribution clearer we have reordered the experiments to highlight the performance improvements first, and only followed by ablation study afterwards. Note that in the revised manuscript we added an “Evaluation” section at the end of Section 2.4 which summarises our experiments, and moved the ablation study at the end of the “Experiments and Results” section (Section 3.5).

---

> > ### Comment · Area_Chair_Db6H · 2026-02-01
> > **Update the final ratings**
> >
> > Please kindly review the authors’ rebuttal and update the final rating by February 1, 2026 (23:59 AoE).

---

### Author Rebuttal · Authors · 2026-01-24

**Rebuttal:**

We thank all the reviewers for their constructive and detailed evaluation of our work. Please find attached the revised manuscript where corresponding edits have been highlighted throughout. Please note that in the revised manuscript the ablation study has now been moved to Section 3.5. We sincerely hope that we have addressed all of the reviewers’ questions and concerns.

**Supporting Material:**

/attachment/4345ab4a340a564dcb9d7531307fb53eb93d5fdc.pdf

---

### Meta-Review · Area_Chair_Db6H · 2026-02-09

**Recommendation:** Accept (Poster)
**Confidence:** 4

**Metareview:**

All reviewers found the proposed method for anatomically consistent fetal cortical reconstruction to be technically sound and the results across multiple datasets highly promising. The addition of sulcal depth guidance effectively ensures the vertex-wise correspondence necessary for developmental studies while maintaining remarkable computational efficiency.

---

### Decision · Program_Chairs · 2026-02-13

Accept (Poster)